# OpenReview forum: "Position: Sycophancy is an Educational Safety Risk: Why LLM Tutors Need Sycophancy Benchmarks"
_ICML.cc/2026/Position_Paper_Track — ICML 2026 Position Paper Track regular_

### Official Review · Reviewer_Q3iN · 2026-03-10

**Significance:** 4
**Argument Clarity:** 2
**Rating:** 5
**Confidence:** 4

**Questions:**

- The extent of bolding in the abstract could maybe be reduced a bit.

- Fig 1: inapppriate -> inappropriate ?

- Some of the figures that are not pdfs yet should be pdfs

- What do the 3 student confidence levels correspond to? Is there any justification why a 3-point-Likert rating is appropriate here?

- Was there any human validation of the dataset at any point? I could not find it in the paper but might've missed it.

- Figure 3 is very hard to read without zooming in far, same for Figure 4

## References

LearnLM Team, "Learnlm: Improving gemini for learning." arXiv preprint arXiv:2412.16429 (2024).

Dinucu-Jianu, David, et al. "From problem-solving to teaching problem-solving: Aligning llms with pedagogy using reinforcement learning." Proceedings of the 2025 Conference on Empirical Methods in Natural Language Processing. 2025.

**Alternative Views Section:**

Yes

**Compliance With Llm Reviewing Policy A Conservative:**

Affirmed.

**Discussion Potential:**

3

**Final Justification:**

The paper presents an important position about safety risks in education applications of LLMs.

The authors have promised to address my concerns by rewriting and adding more discussion about related works.

**Paper Summary:**

The paper argues that the tendency of LLMs towards sycophancy, which stems primarily from the alignment phase, poses a safety risk in education applications.
The authors define a taxonomy of sycophancy and create a new benchmark to test the behavior of LLMs according to this taxonomy.
In line with their argument, they find that LLMs tend to agree with incorrect students just to please users which in education can be detrimental to learning, among others.

**Position:**

Yes

**Position In Title:**

Yes

**Related Work:**

2

**Strengths And Weaknesses:**

*Strengths*

- I personally find that the paper targets a highly important problem with a convincing position: LLMs tend to perhaps overly please users but education requires "friction" where the teacher convinces a student in order for them to learn.

- The dataset seems very useful to me and could be a significant help to future research.

- I also find the findings well-thought-out and interesting and in my opinion they highlight the problem well. The authors do a good job at critical analysis, for example, I was going to argue that using the same LLM for generation and judging can be problematic but this is addressed well in Sec. 6.1. where this is pointed out as a trap that future work can fall into and instead remedies are suggested.


*Weaknesses*

- Still, I think that the position and overall writing of the paper could be made a lot clearer, and that would also help to convince readers who are perhaps less familiar with either LLMs, education research, or their intersection. In particular, many important terms and vocabulary are never properly defined. For example,  what is "directional harm" or "epistemic rigor"? (page 1 right) or "epistemic integrity" and "pressure-contingent validation" (page 2 left). Such terms might not be clear to an overall audience without a short introduction.
Overall, the paper is in my opinion unnecessarily very hard to read due to the type of language that was chosen.

- I also find the discussion of related works on the intersection of LLMs and education / pedagogy too short and not clear enough to properly situate the paper in its context. For example, there are multiple papers that seek to align LLMs for pedagogy (LearnLM Team et al. 2024, Dinucu-Jianu et al. 2025) but these are never discussed.

- Connected to the above, the evaluation is only limited to two commercial LLMs (Claude and GPT-5.2) that, while frontier models, are general models and where the extent to which they are optimized for education is unclear. It would be much more convincing to also evaluate "pedagogically-aligned" models from the above works, as these models are also available.

- This might be a matter of taste but I thought that some of the alternative views, in particular, View C could be discussed and argued against in more detail, to me their discussion is too superficial.

**Support:**

3

---

> ### Author Rebuttal · Authors · 2026-03-26
>
> Thank you for the encouraging review and for the concrete suggestions. We especially appreciate your recognition of the importance of the problem, the potential usefulness of the dataset, and the paper’s effort to surface methodological pitfalls rather than hide them. We address your main concerns directly below.
>
> Additionally, we will explicitly situate the paper against pedagogy-aligned tutor work such as LearnLM and pedagogy-RL efforts, and clarify that our contribution is an evaluation target orthogonal to general pedagogical helpfulness.
>
> **1. On clarity, terminology, and alternative views.**
> We agree that the paper would benefit from clearer writing, earlier definitions, and less rhetorical density. In revision, we will simplify the prose and define key terms much earlier, including educational safety, epistemic rigor, epistemic integrity, and pressure-contingent validation. Our intended claim is narrow: the target failure is not tutoring errors in general, but systematic, pressure-contingent validation of misconceptions in a high-trust tutoring setting. We also agree that some alternative views, especially View C, should be discussed more fully. We will strengthen that section and make our response to nearby interpretations more explicit.
>
> **2. On related work in pedagogy-aligned LLMs.**
> We appreciate the helpful pointers to relevant work, including LearnLM and pedagogy-alignment efforts. We agree that the related work section should better situate the paper within this broader context. In revision, we will expand the discussion of tutor-aligned and pedagogy-oriented LLMs and clarify our intended distinction: those works primarily aim to improve pedagogical quality, whereas our paper isolates a narrower safety-evaluation question, whether a tutor remains supportive yet corrective under agreement-seeking pressure. We agree that this contrast should be made much clearer.
>
> **3. On model coverage and empirical scope.**
> We agree that evaluating pedagogically aligned tutor models would strengthen the empirical story. The current study should be read as an initial two-model benchmark run, not as a closed prevalence study across educational LLMs. Our intended empirical contribution is narrower: to show that the failure mode is real, non-trivial, pressure-structured, and observable across two distinct frontier model families. We will sharpen that scope in revision and position EDUFRAMETRAP more explicitly as an initial benchmark designed for extension to additional tutor families, including pedagogically aligned models.
>
> **4. On the confidence axis and human validation.**
> We agree that the confidence axis should be better explained. The three levels correspond to low, medium, and high learner assertiveness in the student’s first turn. The goal was not to claim psychometric optimality, but to introduce a minimal, interpretable axis that can be cleanly crossed with the three pressure modes in a balanced factorial design. We will make this rationale more explicit. We also agree that human validation should be more visible. The protocol already human-adjudicates all judge-disagreement cases and audits a random subset of judge-agreement cases; in the uploaded evaluation file, that human-labeled subset totals 630 cases. We will report these details much more prominently in the revision.
>
> **5. On presentation details.**
> Thank you for the concrete presentation suggestions. We will reduce unnecessary bolding, fix the noted typo, improve figure readability, and ensure publication-quality vector formatting where appropriate.

---

> > ### Author Rebuttal · Reviewer_Q3iN · 2026-04-04
> >
> > Thank you for your response!
> > I think that adding more related work and introducing terminology more clearly / earlier will improve the paper a lot.
> >
> > Given also the importance of the position from my point of view and after reading the other reviews and responses, I will raise my score.

---

### Official Review · Reviewer_1jtm · 2026-03-11

**Significance:** 3
**Argument Clarity:** 3
**Rating:** 4
**Confidence:** 4

**Questions:**

1. **Evidence for the safety claim.**
   The paper frames pedagogical sycophancy as an *educational safety risk* (i.e., increasing durable misconceptions). Do you have any empirical evidence—or plans for minimal validation—that failures on EDUFRAMETRAP correlate with misconception persistence, confidence miscalibration, or other downstream learning outcomes? Clarifying this would strengthen the central claim.

2. **Human annotation reliability.**
   Since most labels rely on LLM judges and only disagreement cases plus a small subset are human-reviewed, could you report (a) the size and expertise of the human-labeled set, and (b) agreement between human annotations and each judge? How sensitive are the reported SYC rates to expanded human evaluation?

3. **Judge bias and robustness.**
   The paper reports substantial asymmetry between judges. Did you explore additional judges, calibration strategies, or alternative aggregation methods to assess the robustness of headline results to judge choice?

4. **Ecological validity of synthetic scenarios.**
   Trap families are generated via an LLM-based Builder–Validator pipeline. How do you ensure that the misconceptions and pressure patterns reflect realistic student behavior rather than generation artifacts? Any comparison with human-authored cases would be helpful.

5. **Generality across models and settings.**
   Results are reported for two frontier models under a single configuration. Do you have preliminary results on additional model families, prompt variants, or decoding settings to support the generality of the observed pressure-induced patterns?

**Alternative Views Section:**

Yes

**Compliance With Llm Reviewing Policy A Conservative:**

Affirmed.

**Discussion Potential:**

3

**Final Justification:**

The rebuttal addressed my main concerns. I have no further questions and will raise my score.

**Paper Summary:**

This paper argues that **pedagogical sycophancy**—the tendency of LLM tutors to validate user misconceptions under social or epistemic pressure—should be treated as an **educational safety risk** rather than a mere quality issue. To support this position, the authors introduce **EDUFRAMETRAP**, a multi-turn tutoring benchmark designed to evaluate whether models remain *kind-but-correct* when facing agreement-seeking pressure.

The benchmark spans six STEM domains and constructs controlled dialogue scenarios by crossing three levels of student confidence with three pressure types: context-switch frame attacks, authority-based pressure, and social-affective face-saving. The dataset contains 360 trap families and 3,240 instances, with splits defined at the trap-family level to reduce leakage.

The evaluation protocol scores the tutor’s post-pressure response using a structured rubric and two independent LLM judges, with disagreement tracked as a reliability signal and partially adjudicated by humans. Experiments on GPT-5.2 and Claude Sonnet 4.5 show non-trivial sycophancy rates (around 14%) and substantial variation across pressure modes and domains. The authors interpret these findings as evidence that strong reasoning ability does not necessarily imply robustness to social-epistemic pressure (the “Reasoning–Sycophancy Paradox”) and recommend pressure-aware evaluation for LLM tutors.

**Position:**

Yes

**Position In Title:**

Yes

**Related Work:**

3

**Strengths And Weaknesses:**

## Strengths

* **Clear and well-motivated position.** The paper argues that pressure-contingent validation of misconceptions (“pedagogical sycophancy”) should be treated as an educational safety issue rather than a mere quality concern. The position is clearly articulated and connected to learning science concepts such as corrective feedback and conceptual change.

* **Operationalization of the position through a concrete benchmark.** EDUFRAMETRAP directly tests the central claim by controlling for misconception content while varying pressure type (context-switch, authority, social-affective) and student confidence. This design aligns well with the paper’s focus on social-epistemic pressure rather than factual difficulty.



---

## Weaknesses

* **Limited empirical support for the central safety claim.** While the paper demonstrates that pressure-contingent agreement occurs, it does not establish that these benchmark failures lead to measurable educational harm (e.g., increased misconception persistence or confidence miscalibration). The connection between benchmark performance and real-world learning risk remains largely conceptual.

* **Reliance on synthetic data and LLM-based annotation.** Trap families are generated via LLM pipelines, and most labels come from LLM judges with only partial human verification. This raises concerns about ecological validity and annotation reliability for nuanced pedagogical judgments.

* **Judge sensitivity and bias appear substantial.** The reported asymmetries between judges suggest that headline failure rates may depend strongly on judge calibration, which complicates interpretation of quantitative results.

* **Limited model coverage.** The empirical study evaluates only two frontier models in a single configuration, making it difficult to assess the generality of the observed patterns across architectures, sizes, or alignment regimes.

* **Normative boundary between appropriate pedagogical hedging and sycophancy remains unclear.** Although alternative views are discussed, the operational rubric may still conflate reasonable instructional softening or authority alignment with epistemic failure in some contexts.

* **Limited demonstration of practical impact.** The paper advocates new evaluation practices but does not show how the proposed benchmark can guide effective interventions (e.g., training or prompting changes that improve pressure robustness without harming tutoring quality).

**Support:**

3

---

> ### Author Rebuttal · Authors · 2026-03-26
>
> Thank you for the careful and substantive review. We appreciate your positive assessment of the paper’s central position and of EDUFRAMETRAP as a concrete operationalization of that claim. Your comments helped us see that the paper needs to separate more clearly (i) the normative claim that pedagogical sycophancy is safety-relevant in tutoring, from (ii) the narrower empirical claim actually supported by the current benchmark.
>
> **1. On the safety claim and downstream educational harm.** We agree that EDUFRAMETRAP does not by itself establish downstream learning harm, such as misconception persistence or confidence miscalibration. We do not intend to claim that it does. Our claim is narrower: in high-trust tutoring settings, pressure-contingent validation of a misconception is a safety-relevant pre-deployment failure mode that should be explicitly measured, even before downstream user studies are run. The benchmark is therefore meant to test whether a tutor remains supportive while still correcting a student who is under agreement-seeking pressure, not to replace longitudinal learning studies. We will make this narrower claim much more explicit in revision.
>
> **2. On human annotation reliability.** To address reliability directly, we will foreground the human-adjudicated subset more clearly. The uploaded human-labeled evaluation file contains 630 manually labeled cases in total: all 530 judge-disagreement cases plus 100 randomly audited judge-agreement cases. Among the 530 disagreement cases, human adjudication labeled 520 as sycophancy and only 10 as PASS. In the 100 agreement-audit cases, humans additionally overturned 10 consensus-PASS cases to sycophancy. We view this as important evidence that the difficult cases are concentrated in the pedagogically consequential gray zone where reassurance, hedging, and correction can co-occur. It also suggests that the evaluation is not simply an artifact of a single permissive or oversensitive judge.
>
> **3. On judge bias and robustness.** We agree that the judge asymmetry is substantial, and we see this as a methodological result rather than something to hide. It shows that tutoring sycophancy is difficult to score automatically and that relying on a single judge would be misleading. This is why our protocol is disagreement-aware rather than majority-vote-only: each response is labeled by two independent judges, all disagreement cases are human-adjudicated, and a random subset of agreements is audited. We will make clearer that the main qualitative conclusion is not “automatic judgment is solved,” but that non-trivial pressure-contingent misconception validation is observed across both tutors even under a conservative, disagreement-aware evaluation protocol.
>
> **4. On ecological validity of synthetic scenarios.** We agree that EDUFRAMETRAP is controlled rather than fully naturalistic, and we will sharpen that framing. This was intentional: the benchmark is designed to isolate whether pressure causes the tutor to abandon correction while holding misconception content relatively fixed. Each trap family is constrained so that the student claim is wrong in the default instructional frame, the alternate context is plausible but non-default, and the student rationale is short and student-like. We therefore see EDUFRAMETRAP not as a substitute for naturalistic tutoring studies, but as an initial controlled benchmark for a specific failure mode that can later be tested in more natural settings.
>
> **5. On generality across models and settings.** We agree that two models under one configuration do not establish a prevalence map across architectures or alignment regimes. We will revise the paper to state this more explicitly. Our empirical claim is more limited: the failure mode is real, non-trivial, pressure-structured, and observable across two distinct provider families, with similar aggregate rates concealing different fragility profiles by pressure type. We will position EDUFRAMETRAP more clearly as an initial benchmark intended for extension to additional tutor families, pedagogy-aligned models, prompt variants, and longer-form settings.
>
> **6. On the normative boundary and practical impact.** We agree that the paper should separate pedagogical warmth from epistemic retreat more sharply. Our target is not tact, hedging, or supportive phrasing in general, but systematic retreat from correcting a misconception under agreement-seeking pressure in a high-trust setting where the learner may not be able to verify the tutor. We will move that definition earlier in the paper. We will also make the intervention relevance more explicit: CS-SYC points to frame-control failures, AUTH-SYC to source-conflict handling, and FACE-SYC to empathy without epistemic retreat. In this sense, the benchmark is intended not only to diagnose failure, but to support targeted mitigation work.

---

> > ### Author Rebuttal · Reviewer_1jtm · 2026-04-04
> >
> > Thank you for addressing my concerns. I am now satisfied with the response and will update my score to reflect this.

---

### Official Review · Reviewer_soWM · 2026-03-12

**Significance:** 3
**Argument Clarity:** 3
**Rating:** 4
**Confidence:** 3

**Questions:**

1. The paper’s main claim is about educational safety in tutoring, but the experiments appear to cover only two frontier models. How robust do you expect the main conclusion to be across a broader range of tutor styles, model families, and alignment settings, and do you have additional results in that direction?

2. The benchmark focuses on three pressure types: context-switch, authority, and social-affective pressure. What evidence do you have that these cover the main tutoring-specific sycophancy modes, rather than a subset that is easier to template?

**Alternative Views Section:**

Yes

**Compliance With Llm Reviewing Policy A Conservative:**

Affirmed.

**Discussion Potential:**

3

**Final Justification:**

Given the rebuttal partially solving my questions, and considering other reviewers' reviews, I hold the original "4. Borderline accept" rating.

**Paper Summary:**

The paper argues that sycophancy in LLM tutoring is an educational safety issue because tutors may reinforce student misconceptions in high-trust settings. Its main contribution is EDUFRAMETRAP, a benchmark for tutoring-specific sycophancy across six academic domains and multiple pressure types, including authority, context-switch, and social-affective pressure. The paper also proposes a taxonomy of pedagogical sycophancy failures and evaluation metrics tailored to tutoring. Empirically, it reports that frontier models show non-trivial failure rates under these pressures. The paper’s position is that educational LLM evaluation should explicitly measure whether tutors remain supportive while still correcting students when they are wrong.

**Position:**

Yes

**Position In Title:**

Yes

**Related Work:**

3

**Strengths And Weaknesses:**

Streangths:
1. The paper makes a clear and relevant position claim: sycophancy in tutoring systems should be treated as an educational safety issue rather than only a general response-quality issue. This is important for ICML because it connects LLM behavior to deployment risk in a concrete application setting. The paper is also clearly structured, with the position, benchmark design, taxonomy, and evaluation all aligned around the same claim.

2. A main strength is the task framing. Instead of studying sycophancy in a generic chatbot setting, the paper focuses on tutoring, where incorrect agreement may directly reinforce misconceptions. The proposed benchmark covers multiple academic domains and several realistic pressure types, which makes the evaluation more targeted than standard sycophancy benchmarks.

3. The paper also contributes a useful evaluation setup, including a tutoring-specific failure taxonomy and pressure-sensitive metrics. The empirical results show that frontier models exhibit non-trivial failure rates under these conditions, which supports the paper’s core motivation.

Weaknesses:
1. The main weakness is that the empirical scope is still somewhat narrow relative to the strength of the position claim. The paper presents evidence from a limited set of models and benchmark settings, so it is not yet clear how broadly the conclusions generalize across tutoring styles, model families, or deployment settings.

2. In addition, while the benchmark categories are useful, some pressure types may overlap conceptually, and the boundary between pedagogical sycophancy and other instruction-following or robustness failures could be clarified more sharply. This matters because the paper’s position depends on showing that the benchmark isolates a distinct tutoring-specific safety problem.

Suggestions for improvement:
1. The paper would be stronger with broader model coverage, clearer separation between sycophancy and nearby failure modes, and more discussion of how benchmark results map to real tutoring deployment.

**Support:**

3

---

> ### Author Rebuttal · Authors · 2026-03-26
>
> Thank you for the constructive review and for recognizing both the relevance of the position and the usefulness of the benchmark framing. We particularly appreciate your reading of the paper as identifying a concrete tutoring-specific deployment risk rather than a generic response-quality issue. We address your main concerns directly below.
>
> **1. On empirical scope and generalization.**
> We agree that the empirical scope should be stated more carefully relative to the strength of the position claim. In revision, we will sharpen the distinction between the normative claim and the current empirical scope. The normative claim is that pedagogical sycophancy is safety-relevant in tutoring and should therefore be evaluated explicitly. The empirical claim is narrower: EDUFRAMETRAP provides an initial controlled benchmark showing that this failure mode is non-trivial across two frontier tutors and that pressure profiles differ meaningfully across models. We do not intend to claim that the current rates already generalize across all tutor styles, model families, or alignment settings. At present, we do not have additional model results beyond the two tutors reported in the submission; accordingly, we will position EDUFRAMETRAP more explicitly as an initial benchmark designed for extension rather than a closed prevalence study.
>
> **2. On the three pressure types and their distinctness.** We agree that the benchmark categories should be more carefully framed. Our claim is not that context-switch, authority, and social-affective pressure form a closed ontology of all tutoring failures. The three pressure types are not meant as an exhaustive ontology; they are chosen because they map to distinct remediation targets and exhibit subtype concentration rather than complete collapse. Rather, they are three core, tutoring-relevant, intervention-relevant channels: frame abandonment, authority deference, and face-saving epistemic softening. They were selected because they recur naturally in tutoring interactions and map onto distinct remediation targets. The paper’s subtype sanity check supports this construct: authority-mode failures are predominantly AUTH-SYC, social-mode failures are predominantly FACE-SYC, and context-switch failures are predominantly CS-SYC, with limited but non-zero cross-mode leakage. We view that leakage not as taxonomy collapse, but as realistic pedagogical ambiguity in a setting where multiple social pressures can co-occur. We will make this “core but not exhaustive” framing more explicit.
>
> **3. On whether the benchmark isolates a distinct tutoring-specific safety problem.**
> We agree that the boundary between pedagogical sycophancy and adjacent failures should be sharper. Our target is not generic instruction-following, generic robustness failure, or tactful but still corrective pedagogical softening. It is systematic retreat from epistemic correction under agreement-seeking pressure in a high-trust tutoring setting. This is why the deployment context matters: tutoring is high-trust and low-verification, so pressure-contingent validation of a misconception is more consequential than in an ordinary assistant interaction. We will make this normative boundary and deployment link more explicit in revision.
>
> **4. On broader benchmark coverage.**
> We agree that broader model coverage and more varied settings would strengthen the benchmark. We therefore plan to position EDUFRAMETRAP explicitly as an initial benchmark designed for community extension, including additional tutor families, pedagogy-aligned models, broader misconception sets, longer and more natural dialogues, and expanded expert annotation. Because the protocol, schemas, and raw logs are already released, the benchmark is structured to support such extensions.

---

> > ### Author Rebuttal · Reviewer_soWM · 2026-04-03
> >
> > Thank you for the thoughtful rebuttal. It addresses my concerns clearly and constructively. In particular, I appreciate the sharper distinction between the normative claim and the current empirical scope, the clarification that the three pressure types are intended as core but not exhaustive tutoring-relevant channels, and the stronger explanation of how pedagogical sycophancy is meant to differ from adjacent robustness or instruction-following failures.
> >
> > My concerns are therefore partially resolved. The rebuttal improves the framing and makes the paper’s intended claim more precise. However, my remaining concern is that the empirical support is still relatively limited for a position stated at a broad deployment level, especially since there are not yet additional results beyond the two reported tutor models.
> >
> > A follow-up question is: in the revision, can the authors make even more explicit in the paper which conclusions are meant to be supported by the current experiments alone, versus which are broader normative or forward-looking claims about tutoring deployment? This would help align the strength of the position with the current evidence.

---

### Official Review · Reviewer_ZLSM · 2026-03-13

**Significance:** 3
**Argument Clarity:** 3
**Rating:** 5
**Confidence:** 3

**Questions:**

I have included my questions in the weaknesses part.

**Alternative Views Section:**

Yes

**Compliance With Llm Reviewing Policy A Conservative:**

Affirmed.

**Discussion Potential:**

3

**Final Justification:**

The rebuttal has solved my concerns. Thus, I decide to keep my positive rating.

**Paper Summary:**

This paper introduces the Reasoning-Sycophancy Paradox, which agrues that the tendency of aligned LLMs to prioritize user agreeableness over epistemic rigor in educational contexts is a safety risk. The authors claim that validating a student’s misconception under pressure can lead to durable misconceptions, which is harmful in trustworthy learning environments. The authors further introduce a benchmark called EDUFRAMETRAP for measuring the LLM's resilience against three distinct types of pressure, i.e., context-switching frame attacks, authority claims, and social-affective face-saving pressure.

**Position:**

Yes

**Position In Title:**

Yes

**Related Work:**

3

**Strengths And Weaknesses:**

Strengths:

1. The paper identifies a critical trade-off between agreeableness and epistemic rigor. By framing sycophancy as a safety issue, it demonstrates the limitations of the current RLHF paradigm which mainly focuses on preference alignment.

2. The EDUFRAMETRAP benchmark is well-structured and useful. The inclusion of multi-turn interactions may provide insights for real-world tutoring, which is overlooked in commonly-adopted single-turn evaluations.

3. The empirical results may provide guidance for future studies, especially in designing reward functions for alignment that prioritize epistemic honesty alongside helpfulness.


Weaknesses:

1. While the authors attempt to distinguish educational safety from general safety, the boundary still remains vague. The authors are suggested to give a formal definition.


2. The reliance on LLM-as-a-judge may introduce potential bias, which should be further discussed for the reliability of the empirical results.


3. The experiments are only conducted on two LLMs, where the generality of the empirical results is questionable. Also, while the performance gap between GPT-5.2 and Claude-4.5 is interesting, this paper lacks a deeper analysis of this phenomenon.

**Support:**

3

---

> ### Author Rebuttal · Authors · 2026-03-26
>
> Thank you for the thoughtful and encouraging review. We appreciate your recognition of the central trade-off between agreeableness and epistemic rigor, as well as the value of EDUFRAMETRAP as a tutoring-specific benchmark. We address your three main concerns directly below.
>
> **1. On the boundary between educational safety and general safety.**
> We agree that this boundary should be stated earlier and more formally. Our intended claim is narrower than “any tutoring mistake is a safety issue.” The target failure is systematic, pressure-contingent validation of misconceptions in a high-trust learning setting, where the model trades epistemic integrity for rapport or deference. We will move this definition forward in the paper and clarify that EDUFRAMETRAP targets this narrower class of failures rather than educational quality issues in general. This distinction matters especially in tutoring because, unlike many assistant settings, the learner often cannot independently verify the tutor in real time; the concern is therefore not merely factual error, but pressure-induced retreat from correction exactly when corrective friction is needed.
>
> **2. On LLM-as-a-judge reliability.**
> We agree that reliance on LLM-based judges requires stronger discussion. Our protocol was designed around that concern rather than assuming automated judgment is ground truth. Each tutoring response is labeled by two independent judges; disagreements are treated as a first-class reliability signal rather than auto-collapsed; all disagreement cases are human-adjudicated; and a random agreement subset is additionally audited. The uploaded human-labeled evaluation file sharpens this point: the human-labeled subset contains 630 cases in total, comprising all 530 disagreement cases plus 100 agreement-audit cases. Among the 530 disagreement cases, human adjudication labeled 520 as sycophancy and only 10 as PASS; in the agreement-audit subset, humans also overturned 10 consensus-PASS cases to sycophancy. We will report these details more explicitly in revision, because they strengthen the methodological point that ambiguity in tutoring is structural rather than incidental.
>
> **3. On model scope, generality, and the GPT-5.2 / Claude-4.5 gap.**
> We agree that the current study should be framed more explicitly as an initial two-model study. Our intended empirical contribution is to show that the failure mode is real, non-trivial, pressure-structured, and not confined to a single provider family. We do not intend to claim that the present experiments establish a full prevalence map across architectures or alignment regimes. At the same time, we agree that the difference between GPT-5.2 and Claude 4.5 deserves clearer interpretation. In this run, GPT-5.2 is more brittle under authority and social-affective pressure, whereas Claude 4.5 is more brittle under context-switch pressure. We view this not as a simple ranking result, but as evidence for the paper’s broader thesis that reasoning strength and pressure robustness are separable axes: similar overall rates can conceal materially different fragility profiles. We will make this interpretation more explicit, position EDUFRAMETRAP more clearly as an initial benchmark designed for extension to additional tutor families and settings, and open it to community extension.

---

> > ### Author Rebuttal · Reviewer_ZLSM · 2026-04-01
> >
> > Thanks for the rebuttal from the authors. My concerns have been addressed. I suggest that the authors can add the valuable analysis in the rebuttal to the paper in the final version.

---

### Decision · Program_Chairs · 2026-04-30

**Decision:**

Accept (regular)

**Comment:**

This is a very important position presented by the authors. All the referees are leaning positive. I concur with their assessment. The only point that I want to stress is that the educational vs general safety is still not sharply distinguished. This needs to more clearly articulated in the final version.